# Are There Differences in Postural Control and Muscular Activity in Individuals with COPD and with and Without Sarcopenia?

**DOI:** 10.3390/arm93010005

**Published:** 2025-02-18

**Authors:** Walter Sepúlveda-Loyola, Alejandro Álvarez-Bustos, Juan José Valenzuela-Fuenzalida, Carla María Ordinola Ramírez, Carol Saldías Solis, Vanessa Suziane Probst

**Affiliations:** 1Faculty of Health and Social Sciences, Universidad de Las Américas, Santiago 7500975, Chile; 2Biomedical Research Center Network for Frailty and Healthy Ageing (CIBERFES), Institute of Health Carlos III, 28029 Madrid, Spain; aabustos@salud.madrid.org; 3Instituto de Investigación IdiPaz, 28029 Madrid, Spain; 4Departamento de Morfología, Facultad de Medicina, Universidad Andrés Bello, Santiago 8370186, Chile; juan.kine.2015@gmail.com; 5Instituto De Salud Integral Intercultural (ISI), Facultad de Ciencias de la Salud (FACISA), Universidad Nacional Toribio Rodríguez de Mendoza de Amazonas (UNTRM), Chachapoyas 01001, Peru; ordinola3073@gmail.com; 6Escuela de Kinesiología, Facultad de Salud, Universidad Santo Tomás, Temuco 4801127, Chile; carolsaldiasso@santotomas.cl; 7Program of Masters and Doctoral Degree in Rehabilitation Sciences, Londrina State University (UEL) Londrina 86038440, Brazil; vanessaprobst@gmail.com

**Keywords:** COPD, sarcopenia, motor activity, postural balance, electromyography

## Abstract

**Highlights:**

**What are the main findings?**

**What is the implication of the main finding?**

**Abstract:**

Aim: The aim of this study was to compare balance performance and electromyographic activity in individuals with COPD, with and without sarcopenia. Method: Thirty-five patients with COPD were classified with and without sarcopenia according to EWGSOP criteria. Balance was assessed using a force platform under four conditions: standing with feet apart and eyes opened (FHEO), eyes closed (FHEC), on an unstable surface (US), and on one leg (OLS). The surface electromyography activity of lower limb muscles and trunks was recorded. Additionally, the timed up and go test (TUG) and the Brief Balance Evaluation Systems Test (Brief-BESTest) were also utilized. Results: Under the FHEO, FHEC, and US conditions, individuals with sarcopenia demonstrated increased velocities, larger oscillation amplitudes, and greater center of pressure displacements under the US condition (*p* ≤ 0.02). They also showed a higher activation of the scalene, sternocleidomastoid, and abdominal muscles during OLS, along with a reduced activation of the tibialis anterior during OLS and US, and a decreased activation of the vastus medialis during FHEC and US (*p* ≤ 0.04). Furthermore, sarcopenic COPD patients exhibited poorer performance on the TUG and Brief-BESTest compared to their non-sarcopenic counterparts (*p* ≤ 0.02). Conclusions: Individuals with COPD and sarcopenia demonstrated greater instability in both bipedal stances and on unstable surfaces, as well as poorer performance in both dynamic and static balance assessments. Furthermore, these individuals exhibited reduced muscular activation in the lower limbs compared to those without sarcopenia.

## 1. Introduction

Chronic obstructive pulmonary disease (COPD) is a heterogenous lung condition characterized by chronic respiratory symptoms due to abnormalities of the airway and/or alveoli that cause persistent, often progressive, airflow obstruction [1]. COPD is also characterized by extra pulmonary consequences that negatively affect one’s physical function and quality of life [2,3,4,5]. In addition, this disease is one of the main causes of disability and mortality worldwide [1,3,4]. While the primary symptoms of COPD are related to respiratory dysfunction, emerging evidence suggests that COPD can also impact other systems, related to muscle function, balance control, and the risk of falls [6,7,8].

Balance impairment has emerged as an important deficit in individuals with COPD, which is manifested as postural instability, gait disturbance, worse performance in dynamic and static balance tests, and an increased risk of falls [6,7,8]. A previous study conducted on individuals with COPD reported impairments in medial–lateral balance control during static balance tasks, accompanied by an increased activation of respiratory muscles and the gluteus medius [7]. Balance impairment is frequently observed in individuals with COPD who have a previous history of falls, a deficit in functional mobility, or a need for supplementary oxygen [9]. In addition, systematic reviews have reported that muscle weakness, physical inactivity, advanced age, and limited mobility are associated factors contributing to balance impairment in this disease [6,10]. The presence of such factors is also closely related to sarcopenia [11], a geriatric condition defined as the presence of low muscle mass, muscle strength, and physical performance [12].

In individuals with COPD, sarcopenia has garnered particular attention due to its high prevalence and impact on both respiratory and musculoskeletal systems [11,13,14]. Sarcopenia exhibits a high prevalence among individuals with COPD, ranging from 15% to 34% [14]. Its presence is strongly linked to worse physical performance, a worse quality of life, worse pulmonary function, and more health costs associated with hospitalization [15]. Furthermore, a recent study reported that individuals with COPD and sarcopenia exhibit worse balance compared to those without sarcopenia [13]. Although, sarcopenia has been associated as an important factor to balance impairment in populations other than COPD, there is a lack of knowledge about the consequences of sarcopenia on postural control in individuals with this respiratory condition [13,14]. For this reason, the objective of this study was to compare the postural control and patterns of muscular activation in different balance tasks in individuals with COPD with and without sarcopenia. This approach aims to provide a better understanding of the mechanisms underlying balance impairments in this population.

## 2. Materials and Methods

### 2.1. Study Design and Sample

This was a cross-sectional study with a convenience sample of thirty-five individuals with COPD. Patients with COPD (≥55 years old) were diagnosed according to the Global Initiative for Chronic Obstructive Lung Disease (GOLD) [1] using whole-body plethysmography [16]. Patients were recruited from the outpatient unit of the University Hospital of Londrina State University, Brazil. Subjects were excluded if they presented exacerbations during the two weeks before inclusion; a diagnosis of bronchial asthma; the presence of neurological, psychiatric, or musculoskeletal diseases; arthritis; heart failure; alcohol dependence; and the use of antioxidant supplements. This study was approved by the university ethics review board, and all participants provided written informed consent (1.830.048). This study was conducted following the Strengthening the Reporting of Observational Studies in Epidemiology (STROBE) statement: guidelines for reporting observational studies [17].

### 2.2. Sarcopenia Diagnosis

Sarcopenia is defined as the simultaneous presence of low muscle mass and low muscle function (handgrip strength and gait speed), according to the European working group on sarcopenia in older people (EWGSOP) [12]. Muscle mass was measured using a bioelectrical impedance (Biodynamics 310^TM^; Biodynamics Corp., Seattle, WA, USA), including fat-free mass (FFM), the fat-free mass index (FFMI-FFM/height^2^), skeletal muscle mass (SMM), and the skeletal muscle mass index (SMMI-SMM/height^2^). FFM was calculated using the formula of Kyle et al. [18]. FFMI was considered reduced if it was <20.35 kg/m^2^ in men or <14.65 kg/m^2^ in women [19]. Handgrip strength (HGS) was assessed using a hydraulic dynamometer (Jamar Plus + Digital 563213; Lafayette Instrument Company, Lafayette, LA, USA). The participants performed the HGS assessment while seated on a chair, with the arm flexed at a 90-degree angle. They were instructed to perform maximum strength with the dominant hand. The highest value among three attempts was selected for analysis. A one-minute rest was allowed between attempts. Low muscle handgrip strength was defined according to a reference range stratified by age and sex for this Brazilian population [20]. Participants’ 4 m gait speed (4MGS) was evaluated in 4 m. Participants were instructed to walk at their usual gait speed on a 4 m track marked by cones; the time was controlled by a stopwatch and the average speed of two attempts was used for analysis. Low physical performance was defined as 4MGS ≤ 0.8 m/s [17].

### 2.3. Static Balance

Static balance was evaluated using a force platform (BIOMEC400, EMG System of Brazil, Brazil), which is considered a reliable instrument to evaluate stabilographic measures in older populations of individuals with and without COPD [7,21]. Participants were assessed for 4 conditions: (1) bipedal position with eyes opened, which was a control position used to represent center of pressure variables during a non-challenging task; (2) bipedal position with eyes closed, which is used to verify center of pressure variables’ responses when visual inputs are removed; (3) bipedal position on an unstable surface, which simulates surface instabilities using a piece of foam (4 inches; medium density) on a platform; and (4) a one-legged stance, which is considered predictive of falls [22]. The variables analyzed were as follows: the center of pressure displacement area (COP-area); the center of pressure displacement velocity in the anterior–posterior direction (Vel-AP); and the center of pressure displacement velocity in the medial–lateral direction (Vel-ML; amplitude of the movement of COP in antero-posterior direction (Amp-AP) and amplitude of the movement of COP in medial–lateral direction (Amp-ML)). Participants performed two trials of 30 s in each condition, and the average for each condition was used for statistical analysis, following a previous study [7].

### 2.4. Muscular Activation

During static balance evaluation in the 4 conditions, muscular activity was recorded using surface electromyography (Trigno^TM^, Delsys Inc., EUA, Natick, MA, USA) and is presented as root-mean-square values (RMSs). Participants’ skin was properly cleaned with an alcohol swab, and then, wireless electrodes were placed on the following muscles: the tibialis anterior (TA), gastrocnemius (GA), vastus medialis (VM), gluteus medius (GM), erector spinae (ES), rectus abdominis (RA), external intercostal (INT), sternocleidomastoid (SCM), and scalene (SC). The electrodes were positioned on the muscle belly, based on the noninvasive assessment of muscles guidelines [23]. For the external intercostal muscle, the electrode was positioned on the second anterior right intercostal space; for the sternocleidomastoid, it was positioned 5 cm from the right mastoid process; for the scalene, the electrode was positioned in the mid-clavicular line at the level of the cricoid cartilage. The electromyographic signal was obtained with pre-amplified active electrodes (gain: 1000) and filtered in a bandwidth range between 25 and 450 Hz, with a sampling frequency of 2 kHz. To compare muscular activity between the COPD patients and the controls, the electromyographic signals obtained during challenging balance tasks (one-legged stance; bipedal with eyes closed and on an unstable surface) were normalized by the non-challenging task (control position = bipedal with eyes opened) and are presented as relative deltas (%ΔμVRMS), following a previous study [7].

### 2.5. Functional Balance

The Brief Balance Evaluation Systems Test (Brief-BESTest) is a brief version of the Balance Evaluation Systems Test (BESTest) [24]. The Brief-BESTest contains 6 sections: (1) biomechanical constraints, (2) stability limits and verticality, (3) transitions–anticipatory postural adjustments, (4) reactive postural response, (5) sensory orientation, and (6) stability in gait. The maximum score is 24 points, and higher scores indicate better balance performance [24].

### 2.6. Dynamic Balance

The timed up and go test (TUG) was assessed to analyze dynamic balance. Participants were requested to stand up from a standardized chair, walk 3 m, turn and walk back to the chair, and sit down again. The best of two attempts was used for analysis [25].

### 2.7. Pulmonary Function

Pulmonary function was assessed with whole-body plethysmography (Vmax^®^, CareFusion, EUA, San Diego, CA, USA). Measurements were performed according to the American Thoracic Society/European Respiratory Society guidelines [15], and the FEV_1_, FVC, and the FEV_1_/FVC ratio and reference values were those described for the Brazilian population [26].

### 2.8. Comorbidities

The quantity of comorbidities was investigated using the Age-adjusted Charlson comorbidities index (ACCI). ACCI includes 19 medical conditions and was scored using the algorithm proposed by Charlson et al. [27].

### 2.9. Statistical Analysis

Continuous variables were presented as the mean and standard deviation (SD) or median and quartile 1 and 3, and categorical variables are presented as the number and percentage. The parametric distribution of the continuous variables was checked using the Shapiro–Wilk test. Student’s *t*-test and the Mann–Whitney U test were used to compare the continuous variables (for example, muscle strength, muscle mass, balance assessments, and muscular activity) among individuals with COPD, with and without sarcopenia. Comparisons of the categorical variables were performed using the Chi square test. The Pearson test and Spearman’s test were performed to analyze the correlation between variables. Correlation coefficients were interpreted as negligible, weak, moderate, strong, and very strong [28]. Statistical significance was considered as *p* < 0.05, with a 95% confidence interval (95% CI). Statistical analyses were performed using the software IBM SPSS 22 (SPSS Inc., Chicago, IL, USA). In addition, a correlogram (a graph of correlation matrix) was created using GraphPad Prism version 8.0.

## 3. Results

The baseline and demographic characteristics are presented in Table 1. A total of 35 individuals were included, with 13 (37%) classified as sarcopenic and 22 (63%) as non-sarcopenic. No differences were observed in age, gender, pulmonary function, and comorbidities between groups. Individuals with sarcopenia presented lower values of body composition variables (BMI, FFMI, SMI, and abdominal girth) and muscle force (HGS) in comparison with those without sarcopenia (*p* < 0.05 for all comparisons).

Regarding functional and dynamic balance, individuals with COPD, classified as sarcopenic, presented worse performance in Brief-BESTest (*p* = 0.003) and TUG (*p* = 0.038) compared to their counterparts without sarcopenia (Table 1).

A comparison of static balance on a force platform between the sarcopenic and non-sarcopenic COPD patients is presented in Table 2. The sarcopenic individuals exhibited greater postural instability compared to the non-sarcopenic counterparts during bipedal stances with both opened and closed eyes, as well as on an unstable surface. While on an unstable bipedal surface, statistically significant differences were observed in all comparisons (*p* < 0.04), and this was not observed in the rest of the evaluations. With the eyes open, significant differences were obtained in both AP and ML velocity, but only in ML amplitude. The latter finding is the only one that was significantly different in the eyes-closed assessment (3.6 ± 2 vs. 2.4 ± 0.7, *p* = 0.022). Finally, there were no differences in the one-legged stance assessments.

Figure 1 represents the percentage of muscular activation using surface electromyography across the four static positions, according to the presence of sarcopenia. COPD individuals with sarcopenia exhibited a higher activation of scalene, sternocleidomastoid, and abdominal muscles during the one-legged stance and a lower activation of tibialis anterior during both the one-legged stance and the bipedal stance on an unstable surface. In addition, those individuals with sarcopenia presented lower vastus medialis activation during the bipedal stance with the eyes closed and the bipedal stance on an unstable surface (*p* ≤ 0.04 for all).

Figure 2 illustrates the correlations between the force platform variables identified as significant in the “Bipedal on unstable surface” condition from Table 2 (COP-a, Vel-AP, Vel-ML, Amp-AP, and Amp-ML) and the muscle activity of muscles identified as significant for the same condition in Figure 1 (SC, VM, and TA). In individuals with COPD and sarcopenia, negative correlations were observed between the force platform variables and the muscle activity of most evaluated muscles, with correlation coefficients ranging from −0.13 to −0.58. Moderate correlations were specifically identified between COP-a and VM (r = −0.48), Vel-AP, and VM (r = −0.42); Vel-ML and SC (r = −0.53); Amp-ML and VM (r = −0.58); and Amp-ML and SC (r = −0.41). In individuals with COPD, without sarcopenia, a moderate correlation was observed only between Vel-ML and VM (r = 0.44). COP-a showed negative correlations with muscle activity in both sarcopenic and non-sarcopenic COPD individuals (Figure 2A). Vel-AP exhibited negative correlations with TA and VM activity in both groups and with SC only in sarcopenic individuals (with r ranging from −0.14 to −0.42). However, in non-sarcopenic individuals, Vel-AP was positively correlated with SC (Figure 2B). Vel-ML demonstrated negative correlations with muscle activity in TA, VM, and SC in sarcopenic individuals (with r ranging from −0.31 to −0.53), whereas in non-sarcopenic individuals, positive correlations were observed in TA and VM (with r ranging from 0.23 to 0.44) (Figure 2C). Amp-AP showed negative correlations with muscle activity in both groups, except for SC in sarcopenic individuals, where a weak positive correlation was noted (r = 0.12) (Figure 2D). Finally, Amp-ML exhibited positive correlations with TA activity in both sarcopenic and non-sarcopenic individuals (with r ranging from 0.09 to 0.21). Additionally, Amp-ML showed negative correlations with VM and SC activity in COPD individuals, with correlation coefficients ranging from −0.002 to −0.58 (Figure 2D). Correlations between force platform variables and the muscle activity of all evaluated muscles during the four balance positions in individuals with and without sarcopenia are presented in a correlogram as Appendix A. The whole correlation matrix is presented as Appendix A.

Finally, as a secondary analysis, the correlation between BMI, FFMI, SMI, FMI, HGS, 4MGS, TUG, and Brief-BESTest with and the five functional parameters (COP-a, Vel-AP, Vel-ML, Amp-AP, and Amp-ML) in each position (bipedal with eyes opened, bipedal with eyes closed, one-legged stance, and bipedal on an unstable surface) was analyzed. In the bipedal stance with eyes open, BMI, FFMI, SMI, HGS, 4MGS, and Brief-BESTest showed significant correlations with Vel-AP (with r ranging from −0.291 to −0.709) and Vel-ML (with r ranging from −0.315 to −0.640), while TUG was only significant for Amp-AP (r = 0.337; *p* = 0.041). In the bipedal stance with eyes closed, notable correlations with Vel-ML were observed for BMI (r = −0.475; *p* = 0.003); FFMI (r = −0.460; *p* = 0.006); SMI (r = −0.462; *p* = 0.009); and HGS (r = −0.446; *p* = 0.011), with TUG remaining significant only for Amp-AP (r = 0.348; *p* = 0.035). Similarly, in the bipedal stance on an unstable surface, BMI demonstrated relevance with Amp-AP (r = −0.422; *p* = 0.010) and Amp-ML (r = −0.419; *p* = 0.011), as did FFMI with Amp-AP (r = −0.588; *p* = 0.000) and Amp-ML (r = −0.593; *p* = 0.000). On the other hand, HGS was significantly correlated with COP-a (r = −0.377; *p* = 0.033); Amp-AP (r = −0.414; *p* = 0.018); and Amp-ML (r = −0.393; *p* = 0.026). No correlations between variables were observed in the one-legged stance (Appendix A).

## 4. Discussion

The present study aimed to compare the postural control and patterns of muscular activation in different balance tasks in individuals with COPD, with and without sarcopenia. Individuals with COPD who also have sarcopenia demonstrated poorer dynamic and functional balance, as well as diminished performance in static balance tasks. These differences were especially evident in the bipodal position with eyes open and on an unstable surface. When we evaluated whether these differences were justified by different activation patterns of the musculature evaluated by electromyography, we did not observe differences in the bipedal position with eyes open, but we did observe differences on unstable surfaces, where individuals with sarcopenia presented less activation of the scalene, vastus medius, and tibialis anterior musculature. Additionally, this reduced muscle activation was associated with greater postural instability. These results highlight the multifactorial impact of sarcopenia, which not only affects muscle mass and strength but also significantly influences functional mobility, balance, and muscle activity in individuals with COPD.

An indicator of balance impairment is the increased center of pressure displacement area, velocity, and amplitude of movement, which was observed in individuals with COPD and sarcopenia, even in the bipedal position with eyes open. Additionally, differences in muscle activation were more pronounced when the task induced greater instability, as observed during one-legged stance evaluations and bipedal standing on an unstable surface. Under these conditions, several differences in muscle activation were noted in individuals with sarcopenia, particularly in the scalenus, vastus medialis, and tibialis anterior muscles. Furthermore, individuals with sarcopenia demonstrated worse performance in dynamic and functional balance tests, such as the timed up and go test and Brief-BESTest, compared to COPD individuals without sarcopenia. Although previous studies have reported greater impairments in static and dynamic balance among individuals with COPD compared to healthy controls [7,9], this study highlights that balance performance is further compromised in those individuals with sarcopenia. Sarcopenia has been identified as a significant factor contributing to balance instability and an increased risk of falls, primarily due to reduced muscle strength and muscle mass [12,13]. Our findings show that greater balance impairment, expressed as an increased center of pressure displacement area, velocity, and amplitude of movement, was negatively correlated with muscle mass, muscle strength, and physical performance. Similar associations have been previously reported by other authors in individuals with COPD [29]. These functional tests and body composition variables, including handgrip strength, muscle mass, and gait speed, are the diagnostic criteria for sarcopenia [12].

Balance impairment represents an important risk factor for falls and fractures, both of which are associated with an increased risk of mortality in individuals with COPD [29,30]. However, according to our results, an underlying mechanism that could explain balance impairment in individuals with COPD could be the presence of sarcopenia. Independently, sarcopenia has been shown to increase the risk of falls and fractures in older adults [31]. In this sense, a recent study has reported that older adults with sarcopenia present with decreased balance function and increased electromyographic activity of the dominant tibialis anterior muscles and a vulnerability to fatigue of the non-dominant gluteus maximus [32]. Conversely, in our study, sarcopenic older adults displayed a reduced activation of the tibialis anterior during both the one-legged stance and bipedal stance on an unstable surface, suggesting impaired lower limb muscle function. A delay in the latency time for lower limb muscles has been previously observed in individuals with COPD, secondary to peripheral muscle weakness, which could affect the reaction time to postural control [33].

The pathophysiological mechanisms of sarcopenia are not yet well understood. However, several muscle factors have been identified in individuals with sarcopenia [34]. These individuals may present a disfunction in skeletal muscle reparation, with a diminished capacity in regenerative aged muscle, a fact that may explain their inability to maintain skeletal muscle mass [35]. Furthermore, mitochondrial dysfunction may be present in older adults with sarcopenia [35], leading not only to a decrease in muscle mass but also affecting the neuromuscular system, directly impacting muscle quality (defined as the ability to generate force per unit of muscle mass) [35,36,37]. Peripheral neuromuscular function is not the only aspect affected by aging. Age-related changes in the nervous system and motor neurons, such as a reduction in spinal excitability and inhibition, as well as variability in the discharge rates of motor units during submaximal contractions, may contribute to the reduced ability of older adults to perform steady muscle contractions [36,37,38,39,40,41]. According to our results, these changes would directly affect balance and muscle activation.

Despite this, balance exercises are not routinely integrated as part of pulmonary rehabilitation protocols [42]. Our results underscore the clinical implication of incorporating balance training into pulmonary rehabilitation for individuals with COPD, particularly those with sarcopenia, as they present heightened postural instability in static and dynamic balance. Integrating balance training into classical pulmonary rehabilitation programs is feasible and effective, not only for improving one’s balance but also improving one’s aerobic capacity and overall quality of life. Nevertheless, there remain few studies in this field. Additionally, our findings suggest that not all individuals with COPD exhibit compromised balance or postural stability; rather, this tendency is probably more pronounced in those with sarcopenia. Thus, as previous studies have emphasized, early detection of sarcopenia in COPD patients is crucial, not only for identifying functional impairments [10,11] but also for guiding tailored intervention strategies [43]. Although we did not observe significant differences in the age-adjusted Charlson Comorbidity Index between individuals with and without sarcopenia, it is essential to recognize that individuals with a higher burden of comorbidities tend to have greater muscle impairment [44]. This increases their risk of sarcopenia, frailty, and disability [11]. Therefore, incorporating balance training into pulmonary rehabilitation programs could be particularly beneficial for COPD patients with sarcopenia and/or other comorbidities that negatively affect balance control [11,44].

Identifying predictive factors is of great clinical importance, especially for effective prevention, treatment, and rehabilitation [45]. This study acknowledges that sarcopenia affects both functional mobility and muscle strength, enabling healthcare professionals to detect early symptoms and implement preventive interventions. However, there are similarities in the clinical aspects of COPD and sarcopenia [46]. Currently, pulmonary rehabilitation programs focus on identifying respiratory symptoms, airflow limitations, and muscular dysfunction, aiming to develop personalized programs [47] due to the comprehensive assessment of older adults. Our results highlight that COPD sarcopenic patients have a reduced ability to maintain balance, making them more prone to falls and fractures, which is associated with increased mortality in older adults. This underscores the crucial role of incorporating coordination, stability, and balance into comprehensive rehabilitation programs, contributing to improvements in the cardiorespiratory system and the musculoskeletal system, which will lead not only to improving one’s aerobic capacity but also the quality of life of patients. Randomized controlled trials combining pulmonary rehabilitation with balance training have demonstrated clinically significant improvements in balance, exercise capacity, quality of life, muscle strength, physical and mental health, as well as reductions in dyspnea and fall risks among individuals with COPD [48,49,50].

This study has several strengths, such as the number of assessments included in the balance test, as well as the four different positions in which these assessments were evaluated. Additionally, the use of age- and sex-stratified cut-off points for sarcopenia diagnosis in the Brazilian population [20] is also a strength of this research. A force platform, considered the gold standard for measuring static balance, was used [7]. Although force platforms are not commonly used in clinical practice, it is noteworthy that the center of pressure displacement area, velocity, and amplitude of movement were correlated with functional tests and body compositions in this study. This highlights that the functional tests used in clinical practice are associated with the gold standard for static balance, detecting fall risks; balance disorders; and geriatric syndromes, such as sarcopenia or frailty [12]. However, our study has several limitations, including the small sample size and cross-sectional design, which limits the findings to an exploratory study. For example, we did not obtain any significant difference in the one-legged stance evaluation on the force platform. It is possible that this non-significant association was due to a lack of statistical power, since in variables such as COP-a, non-significant differences were observed between sarcopenic vs. non-sarcopenic individuals (15 vs. 8.9). Therefore, larger studies, with a longitudinal follow-up to study the evolution of individuals with COPD based on balance and muscle activation over time and based on the presence of sarcopenia are needed in the future. In addition, evaluating the effects of different types of interventions in individuals with COPD based on the presence or absence of sarcopenia should be a priority for the proper management of this population.

## 5. Conclusions

Individuals with COPD and sarcopenia demonstrated greater instability in both bipedal stances and on unstable surfaces, as well as poorer performance in both dynamic and static balance assessments. Furthermore, these individuals exhibited reduced muscular activation in the lower limbs compared to those without sarcopenia.

## Figures and Tables

**Figure 1 arm-93-00005-f001:**
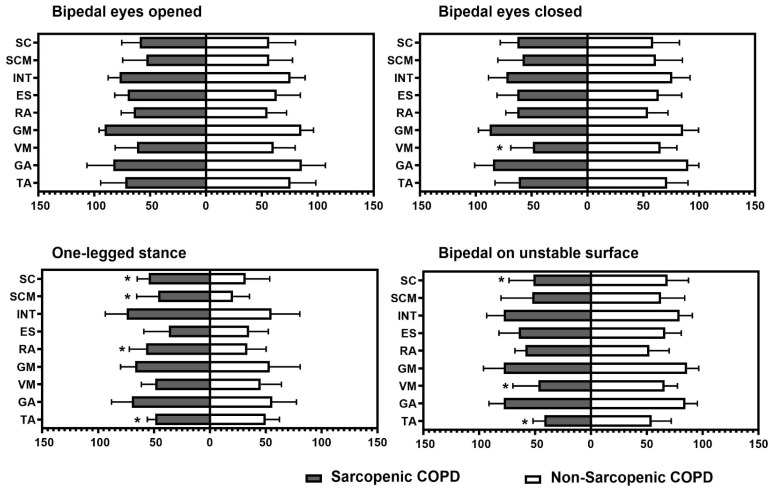
Intergroup comparison of percentage of muscular activation in each task. TA: tibialis anterior; GA: gastrocnemius; VM: vastus medialis; GM: gluteus medius; ES: erector spinae; RA: rectus abdominis; INT: external intercostal; SCM: sternocleidomastoid; SC: scalene. * Statistical significance at *p* < 0.05.

**Figure 2 arm-93-00005-f002:**
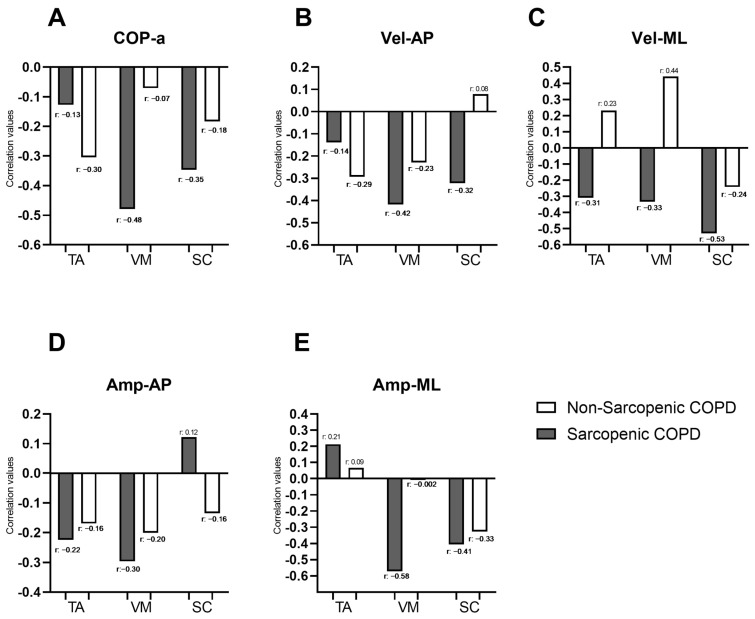
Correlation coefficients between force platform variables (COP.a; Vel-AP; Vel-ML; Amp-AP and Amp-ML; represented by the letters (**A**–**E**), respectively) and the percentage of muscle activation during the bipedal stance on an unstable surface in individuals with and without sarcopenia. COP-area: center of pressure displacement area; Vel-AP: center of pressure displacement velocity in anterior–posterior direction; Vel-ML: center of pressure displacement velocity in medial–lateral direction; Amp-AP: amplitude of the movement of COP in antero-posterior direction; Amp-ML: amplitude of the movement of COP in medial–lateral direction; TA: tibialis anterior, VM: vastus medialis; SC: scalene.

**Table 1 arm-93-00005-t001:** Comparison of characteristics between sarcopenic and non-sarcopenic COPD patients.

	Sarcopenic (*n* = 13)	Non-Sarcopenic (*n* = 22)	*p*
Anthropometric data			
Age (years)	71 ± 7	70 ± 5	0.3
Female, *n* (%)	5 (38%)	7 (32%)	0.93
BMI (kg/m^2^)	20.3 ± 3.7	28.4 ± 3.6	0.001 *
FFMI (kg/m^2^)	13.5 ± 5.7	17.6 ± 1.8	0.01 *
SMI (kg/m^2^)	7.3 ± 1.4	10 ± 1.8	0.004 *
FMI (kg/m^2^)	9.1 ± 4.5	11 ± 2.5	0.15
Abdominal girth	89 ± 9	102.6 ± 10.9	0.006 *
Pulmonary Function			
FVC (L)	2.7 ± 0.6	2.8 ± 0.8	0.8
FVC (%pred)	88 ± 15	80 ± 18	0.3
FEV_1_ (L)	1.2 ± 0.4	1.4 ± 0.5	0.4
FEV_1_ (%pred)	47 ± 10	49 ± 14.2	0.5
FEV_1_/FVC (%)	43 ± 9	49 ± 9	0.06
Comorbidities			
ACCI (score)	4.9 ± 1.2	5 ± 1.3	0.89
Muscle Force			
HGS (kg)	19.8 ± 5.9	24.5 ± 7.4	0.001 *
Gait Speed			
4MGS (m/s)	1.04 ± 0.24	1.2 ± 0.11	0.37
Functional Balance			
Brief-BESTest (score)	15 ± 2	18 ± 3	0.003 *
Dynamic Balance			
TUG (sec)	8.3 ± 1.3	6.9 ± 1.3	0.038 *

The values are described as the mean ± SD. ACCI: Age-adjusted Charlson comorbidities index; Brief-BESTest: Brief Balance Evaluation Systems Test; BMI: body mass index; COPD: chronic obstructive pulmonary disease. FFMI: fat-free mass index; FEV1: final expiratory volume in 1 s; FVC: forced vital capacity; FMI: fat mass index; HGS: handgrip strength; TUG: timed up and go test; SMI: skeletal muscle mass index; 4MGS: 4 m gait speed. * Statistical significance at *p* < 0.05.

**Table 2 arm-93-00005-t002:** Comparison of static balance on force platform between sarcopenic and non-sarcopenic COPD patients.

Variables	Sarcopenic (*n* = 13)	Non-Sarcopenic (*n* = 22)	*p*
Bipedal with eyes opened			
COP-a (cm^2^)	2.4 ± 2	1.3 ± 0.5	0.06
Vel-AP (cm/s)	2.7 ± 0.7	1.9 ± 0.6	0.001 *
Vel-ML (cm/s)	2.7 ± 0.4	1.9 ± 0.6	0.003 *
Amp-AP (cm)	1.5 ± 0.4	1.5 ± 0.5	0.9
Amp-ML (cm)	2.9 ± 1.5	2.1 ± 0.6	0.02 *
Bipedal with eyes closed			
COP-a (cm^2^)	4 ± 5	1.5 ± 0.6	0.1
Vel-AP (cm/s)	2.8 ± 0.7	3.8 ± 4.5	0.6
Vel-ML (cm/s)	2.9 ± 0.6	2.3 ± 0.9	0.007 *
Amp-AP (cm)	2 ± 0.9	1.6 ± 0.5	0.17
Amp-ML (cm)	3.6 ± 2	2.4 ± 0.7	0.022 *
One-legged stance			
COP-a (cm^2^)	15 [10–16.8]	8.9 [7–12.9] †	0.20
Vel-AP (cm/s)	6.7 ± 1.6	5.6 ± 1.6 †	0.27
Vel-ML (cm/s)	6.09 ± 1.7	5.4 ± 1.7 †	0.38
Amp-AP (cm)	4 ± 0.6	4.7 ± 1.1 †	0.9
Amp-ML (cm)	4.9 ± 1	5.3 ± 1.2 †	0.5
Bipedal on unstable surface			
COP-a (cm^2^)	12.6 ± 2.5	3.9 ± 1.78	0.01 *
Vel-AP (cm/s)	3.2 [3–3.6]	2.2 [1.7–3]	0.035 *
Vel-ML (cm/s)	3.5 [3.1–3.5]	2.2 [2–2.4]	0.039 *
Amp-AP (cm)	4 ± 1.4	2.7 ± 0.7	0.003 *
Amp-ML (cm)	5.13 ± 2.1	3.2 ± 0.6	0.001 *

COP-area: center of pressure displacement area; Vel-AP: center of pressure displacement velocity in anterior–posterior direction; Vel-ML: center of pressure displacement velocity in medial–lateral direction; Amp-AP: amplitude of the movement of COP in antero-posterior direction; Amp-ML: amplitude of the movement of COP in medial–lateral direction. † Analysis was performed with *n* = 20. * Statistical significance at *p* < 0.05.

## Data Availability

Data are contained within the article and Appendix A.

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
