# Peer review of "Are There Differences in Postural Control and Muscular Activity in Individuals with COPD and with and Without Sarcopenia?"

_arm, 2025, doi:10.3390/arm93010005_

Round 1

Reviewer 1 Report

Comments and Suggestions for Authors

I would like to thank the editor for the opportunity to review this interesting study.

My comments:

1.      The authors' affiliations must be in order. After 1, 4 currently follows, but it should be 2 and 3.

2.      Update the reference for the GOLD guideline regarding the definition of COPD. The authors are using the 2013 version, but it is updated annually, and we are now on the 2024 edition.

3.      Please update the references. For example, the authors use a 2015 systematic review (ref 6) to discuss postural balance issues in people with COPD; however, there are several reviews on this topic published after 2020.

4.      In the introduction, the authors describe how sarcopenia affects people with COPD very briefly and generally. Since the title includes the concept of "muscular activity," I would expect the text to provide more detailed information regarding the muscular activity of patients with COPD.

5.      Please review the definition of COPD in the Methods section. Paper 14 is not the GOLD guideline.

6.      Were subjects with musculoskeletal issues excluded?

7.      The authors use the 2010 definition of sarcopenia; however, there is an updated version from 2019 (https://pmc.ncbi.nlm.nih.gov/articles/PMC6322506/). Is there any reason why the updated version was not used?

8.      Could the authors provide more detail regarding this assessment: "The highest value from three repetitions (1-minute rest each) was used"?

9.      While I believe that sarcopenia may be related to falls, I do not agree with the authors' assertion: "according to our results, an underlying mechanism that could explain falls in individuals with COPD could be the presence of sarcopenia." This is especially concerning given that this study did not evaluate falls or mechanisms related to them.

10.  Although the objective of this study was not to implement a rehabilitation program, I think it is a great idea to link it with rehabilitation. However, the paragraph is described in terms of the importance of integrating balance exercises, and I believe it should emphasize their effectiveness rather than just feasibility. Specifically, it should demonstrate with evidence that balance training improves specific outcomes.

11.  Undoubtedly, comorbidities play an important role in both sarcopenia and COPD. The authors measured them but do not mention them. I believe it is necessary to add a paragraph discussing comorbidities.

Author Response

Dear Reviewer,

Thank you very much for reviewing the manuscript. Your comments and suggestions have been invaluable in improving the quality of the manuscript. We have implemented the suggested changes and highlighted them in red for easy identification.

Additionally, we have attached a document with a point-by-point response to your comments.

Please let us know if any further revisions are needed.

Kind Regards

Reviewer 2 Report

Comments and Suggestions for Authors

The authors focus on the relationship between postural balance and muscle activity in sarcopenic COPD patients, highlighting the impact of differences in muscle activity on functional abilities, including postural maintenance. This is a highly intriguing study. However, I would like to request the following considerations:

   Clarity of Comparisons in Fig. 2

The comparisons presented in Fig. 2 appear somewhat unclear. It might be helpful to narrow the focus to the items identified as significant only in “Bipedal on unstable surface” in Table 2 (COP-a, Amp-AP) and those significant in Fig. 1 for “Bipedal on unstable surface” (SC, VM, TA). By concentrating on these parameters, the significance of “Bipedal on unstable surface” in sarcopenia could become more apparent.

   Visualization of Key Results in Fig. 2

I suggest re-presenting the key findings from Fig. 2 using bar graphs or similar visualizations. This would enhance the clarity and accessibility of the results for readers.

   Analysis of Table 1 Variables and Functional Parameters

Could you analyze the relationship between the variables listed in Table 1 (e.g., BMI, FFMI, SMI, FMI, Brief-BEST test, TUG test, and HGS) and the five functional parameters (e.g., COP-a, Vel-AP)? Such an analysis would help clarify the extent to which muscle mass influences functionality, thereby adding further credibility to your findings.

Author Response

Dear Reviewer,

Thank you very much for reviewing the manuscript. Your comments and suggestions have been invaluable in improving the quality of the manuscript. We have made the suggested changes and highlighted them in red for easier identification. Additionally, we have attached the tables containing the new analyses performed.

Please let us know if any further revisions are needed.

Additionally, we have attached a document with a point-by-point response to your comments.

Sincerely,

Round 2

Reviewer 1 Report

Comments and Suggestions for Authors

Congratulations to the authors. They have satisfactorily addressed all my comments.

Author Response

Dear Reviewer, we sincerely thank you for reviewing our article and for the constructive comments and suggestions provided

Kind Regards

Reviewer 2 Report

Comments and Suggestions for Authors

The attached previous questions 1 and 2 have not been clearly addressed. Additionally, the discussion of the results in response to question 3 has not been included, making the revision insufficient. In its current state, it cannot be considered acceptable for acceptance.

1-            Clarity of Comparisons in Fig. 2

The comparisons presented in Fig. 2 appear somewhat unclear. It might be helpful to narrow the focus to the items identified as significant only in “Bipedal on unstable surface” in Table 2 (COP-a, Amp-AP) and those significant in Fig. 1 for “Bipedal on unstable surface” (SC, VM, TA). By concentrating on these parameters, the significance of “Bipedal on unstable surface” in sarcopenia could become more apparent.

2-            Visualization of Key Results in Fig. 2

I suggest re-presenting the key findings from Fig. 2 using bar graphs or similar visualizations. This would enhance the clarity and accessibility of the results for readers.

3-         Analysis of Table 1 Variables and Functional Parameters

Could you analyze the relationship between the variables listed in Table 1 (e.g., BMI, FFMI, SMI, FMI, Brief-BEST test, TUG test, and HGS) and the five functional parameters (e.g., COP-a, Vel-AP)? Such an analysis would help clarify the extent to which muscle mass influences functionality, thereby adding further credibility to your findings.

Author Response

Dear Reviewer,

We sincerely thank you for your valuable suggestions and insightful comments. We have carefully revised the manuscript and addressed each of your points in detail. Please refer to the updated version, which incorporates the requested changes.

We are resubmitting the revised manuscript along with a detailed "point-by-point response," where we have comprehensively addressed all your comments. The manuscript has been thoroughly reviewed and approved by all co-authors.

We believe this revised version represents a significant improvement over the previous submission, and we hope it meets the standards required for acceptance.

Kind regards,

Round 3

Reviewer 2 Report

Comments and Suggestions for Authors

The revised manuscript is well written.